# Stress and Mental Health among Children/Adolescents, Their Parents, and Young Adults during the First COVID-19 Lockdown in Switzerland

**DOI:** 10.3390/ijerph18094668

**Published:** 2021-04-27

**Authors:** Meichun Mohler-Kuo, Shota Dzemaili, Simon Foster, Laura Werlen, Susanne Walitza

**Affiliations:** 1La Source, School of Nursing Sciences, HES-SO University of Applied Sciences and Arts of Western Switzerland, 1004 Lausanne, Switzerland; s.dzemaili@ecolelasource.ch; 2Department of Child and Adolescent Psychiatry and Psychotherapy, University Hospital of Psychiatry Zurich, University of Zurich, 8032 Zurich, Switzerland; simon.foster@kjpd.uzh.ch (S.F.); susanne.walitza@pukzh.ch (S.W.); 3Department of Clinical Research, University Hospital Basel, University Basel, 4031 Basel, Switzerland; Laura.Werlen@usb.ch

**Keywords:** children, adolescents, mental health, stress, internet use, COVID-19, psychological impact, youth, pandemic

## Abstract

The present study aimed to assess various stressful situations and the psychological impact of the first COVID-19 pandemic lockdown among youths in Switzerland. We included samples of 1627 young adults aged 19–24 from the Swiss Youth Epidemiological Study on Mental Health and 1146 children and adolescents aged 12–17 years and their parents. We assessed symptoms of various mental health problems, internet use, and perceived stress during the first COVID-19 lockdown. In the analyses, data were weighted to be representative of the Swiss population. During the first lockdown in Switzerland, the most common sources of perceived stress were the disruption of social life and important activities, uncertainty about how long the state of affairs would last, and the pandemic itself. In addition, around one-fifth of the young adults met the criteria for at least one of the mental health problems (attention deficit hyperactivity disorder, depression, generalized anxiety disorder), while one-third of children/adolescents screened positive for at least one of the mental health problems (attention deficit hyperactivity disorder, oppositional defiant disorder, depression, anxiety). Moreover, 30.1% of children and 21.3% of young adults met the criteria for problematic internet use. The study showed considerable stress perceived by young adults and symptoms of mental health problems, especially among females, during the first COVID-19 lockdown in Switzerland.

## 1. Introduction

Since the first case of coronavirus disease 2019 (COVID-19) was reported in early December 2019, the world has been struggling with its ever-increasing spread, which was declared a pandemic by the World Health Organization on 11 March 2020. The outbreak of infectious diseases like COVID-19 can be expected to have a psychological impact on the well-being of the general population due to fear of the disease itself and social isolation due to containment measures [1]. Many countries have been forced to implement extreme measures, such as quarantines, social distancing, or in some cases, total lockdown to prevent the collapse of the health system due to the rapid spread of the disease. Such social isolation and distancing have a significant impact socially, economically, and psychologically. Like adults, youths also have to face drastic changes in lifestyle, including perpetual termination of schooling, fear of being infected or spreading the virus to vulnerable family members, everlasting ennui, frustration, lack of in-person contact with peers and teachers, lack of personal space at home, family isolation if institutionalized, or even an increase in domestic violence and abuse [2]. Such stressful situations are generally likely to cause mental health problems in turn [3,4,5,6]. It is important to monitor the prevalence of mental disorders in order to identify the pandemic’s impact on mental well-being.

In general, mental health problems in children and adolescents are especially relevant because they may have long-term effects later in life [3,4,6]. Compared to adults, young people are more vulnerable to their immediate environment and have fewer resources and past experiences to cope with stressful situations [7]. Therefore, they are more affected by stressful situations due to feelings of uncertainty. The extent and ability to which children and adolescents can cope are subject to individual and familial or parental-related risk and resilience factors [7]. However, whereas substantial research on the COVID-19 pandemic’s impact on the mental health of adults has accumulated [5,8,9], less is known about its impact on children and adolescents [8,10]. A recent systematic review identified three studies that reported prevalence rates of depression among children and adolescents ranging from 22.6 to 43.7% [11]. The same review identified two studies reporting prevalence rates for the presence of anxiety symptoms at 18.9% and 37.4%, respectively. The review also reported prevalence rates of specific emotional reactions, such as “fear for the health of relatives” (22.0%), concerns “about health and life threats posed by COVID-19” (40.0%), and being “moderate[ly]/quite worried about being infected with the virus” (62.2%). However, most of the studies used snowball or convenience cluster sampling with small sample sizes. In addition, a majority of the studies were from China, which has experienced vastly different circumstances with regard to the COVID-19 pandemic due to disparate approaches to regulating lockdowns and distinct cultural attitudes.

Although data on youths have started to accumulate, relevant data for Switzerland are lacking. Like many other countries, Switzerland has been forced to implement extreme measures and enacted a total lockdown. On 16 March 2020, schools and most shops were closed nationwide, and on 20 March, all gatherings of more than five people in public spaces were banned. From 17 March onward, measures such as quarantines, social distancing, or in some cases, total lockdown were implemented to prevent the collapse of the health system due to the rapid spread of the disease. A gradual opening began at the end of April until a total opening on 8 June 2020. The present study aimed to assess the impact of this first lockdown due to COVID-19 by reporting the prevalence of symptoms of mental illness among children, adolescents, and young adults from nationally representative samples of these populations and to describe various stressful situations due to COVID-19 perceived by youths and parents.

## 2. Materials and Methods

### 2.1. Study Samples

The present study included three different samples: (1) young adults from the Swiss Youth Epidemiological Study on Mental Health (S-YESMH), (2) children and adolescents ages 12–17 years old, and (3) the parents of children and adolescents from sample 2.

#### 2.1.1. Young Adult Sample

We resurveyed young adults from S-YESMH in order to assess the change in their mental health in the time before and after the COVID-19 pandemic lockdown. S-YESMH is a nationally representative observational study of Swiss young adults’ mental health and wellbeing that was funded by the Swiss National Science Foundation. S-YESMH was first conducted in 2018. In the first wave, 3840 young adults born from 1996 to 2000 participated in the survey. Details on the survey have been published elsewhere [12].

We worked to update all participant addresses using the Swiss postal service address system. Out of the original 3840 addresses, 3654 were still available in the system. A letter of invitation followed by two reminders was sent to the participants from July to September 2020. This letter described the study, stated that the survey was voluntary and that answers would never be linked to participant contact information, and provided free hotline numbers for participants to contact the study team and the market research organization LINK Institute (www.link.ch accessed on 28 February 2021), which collected the data via an online platform. Participants could access the online platform by entering the web address and password provided or by scanning an individualized QR code on the invitation letter. The survey included questions regarding perceived stress related to the COVID-19 pandemic, mental health status, alcohol and internet use, and coping strategies. By mid-October 2020, 1627 young adults had participated in the survey (response rate = 44.4%).

#### 2.1.2. Samples of Children and Adolescents and Their Parents

We aimed to obtain a representative community sample of 1000 children and adolescents that were 12–17 years old from all three language regions (German-, French-, and Italian-speaking) in Switzerland. To recruit children and adolescents, it was necessary to obtain informed consent from both the children and their parents. To reach our goal, we collaborated with the LINK Institute and recruited parents with children 12–17 years old through the LINK Internet Panel. This panel includes more than 100,000 subjects that are representative of the internet-using Swiss population aged 15 to 79 years. All participants in the LINK Internet Panel were recruited within the framework of representative telephone studies, which guaranteed a representative sample.

A letter of invitation describing the study was sent by email with a link to the survey platform to potential participants (parents) in all three language regions. At the beginning of the survey, the potential participants were asked (1) whether they agreed to participate in the survey, (2) whether they had children that were 12–17 years old in their household, and (3) whether they allowed their children to participate in the survey. If all three questions were answered with “yes,” the parents started the survey, which lasted around 15 min. The questions covered their mental health status, stress, and the impact related to the COVID-19 pandemic and lockdown. After the parents completed the survey, their children could continue the survey right away or complete the survey later. In total, 1266 parents and 1150 children completed the survey. We analyzed only the parent–child pairs for which both the parent and child questionnaires were completed. In addition, we weighted our analyses of the children’s data based on their gender, age, and seven major Swiss regions such that these children were representative of the Swiss population. Four children did not have complete data for the weights and were therefore excluded. The final analysis set included 1146 parents and children respondents.

#### 2.1.3. Ethical Considerations

The survey was voluntary and all respondents agreed to participate by beginning the survey. The consent for the children and adolescent sample was obtained by both the parents and children participants themselves. The data collection and data handling procedure strictly complied with the EU’s General Data Protection Regulation (GDPR) and the Helsinki Declaration and was approved by the Ethical Committee in Canton Zurich (2020-01736) and Canton Vaud (2020-01736_2007). For the S-YESMH sample, all of the parties involved in data collection and handling, including the LINK Institute, had to sign a contract with the Swiss Federal Office of Statistics to meet all of the requirements and regulations on data protection of personal information.

### 2.2. Measures

The questionnaire used for young adults was based on the S-YESMH survey from 2018. We used the same questionnaire to assess mental health status and substance use [12]. Additionally, we added some questions to assess the impact of the COVID-19 pandemic on school, work, or family life; perceived stress due to the COVID-19 pandemic; coping strategies; internet use. We used this same modified questionnaire for the parent sample. The questionnaire for children and adolescents was developed in collaboration with the CORONA HEALTH APP study in Germany. The app for this study was supported by the Robert Koch Institute, and the children’s questions were developed by the Department of Child and Adolescent Psychiatry and Psychotherapy at University Hospital Würzburg [13]. We used the same questions as this study did to assess general and mental health, as well as the impact of COVID-19 on school, family, and work. Additionally, we added some questions on internet use, perceived stress due to the COVID-19 pandemic, and coping strategies. All of the questionnaires were translated into German, French, and Italian.

#### 2.2.1. Symptoms of Mental Health Problems

(1)Young adults and parents

We assessed the self-reported symptoms of mental health problems, including depression, generalized anxiety disorder (GAD), and attention deficit hyperactivity disorder (ADHD), using the same instruments in the S-YESMH questionnaire from 2018, which allowed us to compare the mental health status before and after the COVID-19 pandemic. Two Patient Health Questionnaire (PHQ) screeners were used to assess the symptoms of mental health problems: the Generalized Anxiety Disorder 7 (GAD-7) for GAD [14] and the PHQ-9 for depression [15]. Both are widely used in clinical settings and have been validated in populations across the world [16,17,18]. We asked about symptoms of both anxiety and depression during the COVID-19 lockdown using a four-point rating scale for which 0 indicates “not at all” and 3 indicates “always.” Based on the literature, we categorized the total scores into “none,” “minimal,” “mild,” “moderate,” and “moderately severe and severe.” ADHD-related symptoms were assessed using the Adult ADHD Self-Report Scale Screener (ASRS-v1.1), which is a validated six-item instrument about symptoms of ADHD during the past 6 months [19,20,21]. This instrument has adequate sensitivity (68.7%) and is highly specific (99.5%) [19]. We dichotomized the total scores into “no ADHD” (scores below 14) and “ADHD” (scores 14–24) [22]. We also assessed which participants had at least one of these mental health problems.

(2)Children and adolescents

To maximize the response rate, we kept the questionnaire duration under 20 min, selecting screening questionnaires that were short, yet well established. We used the same questionnaires as in the CORONA HEALTH APP study in Germany to assess the symptoms of ADHD, oppositional defiant disorder (ODD), depression, and anxiety. ADHD- (four items) and ODD- (three items) related symptoms were assessed using the screening questions from the Kiddie Schedule for Affective Disorders and Schizophrenia (K-SADS) [23]. These items were assessed with responses of 1 (never), 2 (subthreshold—sometimes, but it does not cause a problem), or 3 (yes, and it causes a problem for me). If one of the four ADHD items or one of the three ODD items were scored 3, the participant screened positive. These screening items have been shown to validly identify children with ADHD and ODD [24].

Anxiety symptoms were assessed using the brief version of the well-established Spence Children’s Anxiety Scale for Children (SCAS-C) [25]. The brief version is comparable with the full version of the SCAS-C and has good validity and reliability. These items assess children’s anxiety symptoms using a four-point scale (never (0), sometimes (1), often (2), and always (3)). We calculated a total score by summing the eight items and created a dichotomized variable using the cutoffs of 7.5 for girls and 5.5 for boys, as recommended by the literature [25]. Furthermore, we used three questions from the PHQ to assess depression (PHQ-2, two questions) and sleep problems (one question). The PHQ-2 has been shown to be an effective measure for depression screening. The sum scores of the two items were further dichotomized as positive vs. negative with a cutoff point of 3 [26].

#### 2.2.2. Internet Use

We asked young adults, children, and adolescents about their average internet use time per day during the COVID-19 lockdown. In addition, we assessed problematic internet use using the eight-item short form of the compulsive internet use scale [27]. The scale consists of eight items with five dimensions of compulsive internet use: conflict, coping, loss of control, preoccupation, and withdrawal symptoms. These symptoms were assessed using a five-point Likert-type scale from 0 (never) to 4 (very often). We dichotomized the total scores into “problematic internet use” (≥13) vs. none (<13) according to the literature [27].

#### 2.2.3. Perceived Stress

Perceived stress was measured using the first part of the Responses to Stress Questionnaire (RSQ)–(COVID-19). The RSQ includes different versions for children/adolescents, parents, and adults and was developed by the Stress and Coping Research Lab at Vanderbilt University [28]. The first part of the questionnaire includes a checklist of 14 situations about COVID-19 that respondents sometimes find stressful or have a problem dealing with. The respondents rate the specific situation in terms of how often each stress has occurred during the lockdown period of the COVID-19 pandemic using a four-point scale (“not at all” (1), “a little” (2), “somewhat” (3), and “very” (4)). We reported the percentage of participants who answered “somewhat” or “very.”

### 2.3. Data Analysis

All statistical analyses were conducted using R version 4.0.3 [29] and STATA version 16 [30]. For young adults, we adjusted our estimates to account for the original stratified sampling design described above and non-responses using the R package “survey” [31]. To examine the potential bias caused by non-responses, we compared the mental health outcomes from 2018 between respondents and non-respondents in the current survey. We found a slightly higher proportion of those who had mental disorder symptoms in 2018 among the non-respondents. In particular, participants with symptoms of depression were less likely to participate in the second survey. In addition, males and non-Swiss participants in the 2018 survey were also less likely to respond in the present survey.

To report results that are representative of all children in Switzerland, we weighted the data by age, gender, and the seven major regions in Switzerland. It should be noted that non-Swiss residents are underrepresented in our sample, and we were not able to consider this variable in the weighting because some strata did not have non-Swiss participants.

Weights were calculated based on the children’s characteristics (age, gender, and region). Therefore, it was not possible to apply these weights to parents’ data. To take the sex differences in mental health outcomes among the parents into consideration, we reported results separately by gender. All analyses, including standard errors, 95.0% confidence intervals, and Wald statistics, for the children were weighted using the STATA survey estimation procedure [30], which used the Taylor series methods available in the STATA survey options to reflect the sampling design.

We used contingency tables to present the prevalence symptoms of mental health outcomes, media use, and perceived stress by gender. A statistically significant difference by gender was indicated if the two confidence internals did not overlap.

## 3. Results

Among the 1627 young adults [surveyed, 60.4% were female, with a mean age of around 21.5 years old (SD = 1.7, range 19–24). The mean age for the 1146 children was 14.5 (SD = 1.7, range 12–17), and most child respondents were Swiss (94.0%). For each child respondent, one parent also responded to the survey as well, two-thirds of whom were the children’s mothers. The mean age of the parents was 47 (SD = 4.7, range 32–55). Other sociodemographic characteristics of the respondents are shown in Table 1.

### 3.1. Perceived Stress Due to COVID-19

Table 2 shows the percentage of parents, children/adolescents, and young adults feeling “rather” or “very” stressed about various situations related to COVID-19. The highest perceived stress was attributed to the disruption of social life and important activities, uncertainty about how long the current state of affairs will last, and the pandemic itself (i.e., infection with the virus), along with distressing news about the pandemic. In particular, the mothers and female young adults felt most stressed by “not being sure about when COVID-19 will end” (41.0% and 53.9% respectively) and being “unable to spend time in person with friends and family” (38.5% and 55.2% respectively), whereas fathers and male young adults felt most stressed by “having to change, postpone, or cancel important plans or events” (33.5% and 44.1% respectively) followed by “not being sure about when COVID-19 will end” (28.0%) for fathers and “unable to spend time in person with friends and family” (43.3%) for male young adults. Children and adolescents felt most stressed by “having to change, postpone, or cancel important plans or events” (48.1% for girls and 35.1% for boys) and being “unable to participate in social activities and normal routines” (41.0% for girls and 37.9% for boys). On the other hand, stress due to financial problems (6.7–12.6%), trouble getting groceries and other needed supplies (2.6–8.4%), and medical care and mental health services (6.9–15.4%) was somewhat less prevalent.

Despite the similarities among the samples, there were also apparent differences. For example, across all samples, females reported higher (sometimes substantially higher) stress than males. In general, stress related to “spending time with friends or family; participating in social activities and normal routines; having to change, postpone, or cancel important plans and events” was more prevalent among young adults and children as compared to parents. Stress due to difficulties completing school or work responsibilities online was highest among children (16.5% in boys, 18.7% in girls). On the other hand, higher proportions of young adults (male: 17.8%, female: 24.1%) were stressed by being unable to complete educational or work requirements than the other samples. The largest proportion of subjects stressed by the need to “take on greater family or work responsibilities” was found among mothers (23.0%, vs. 12.3% among fathers). While there were differences in the proportions of subjects experiencing stress among the groups, it should be noted that within groups, the rank order of the stressful situations’ prevalence was remarkably similar, suggesting that all subjects identified the same areas as being the most stressful during the pandemic.

### 3.2. Mental Health Problems

We compared the results for the 1627 young adults who participated in S-YESMH in both 2018 and the present survey (Table 3). During the lockdown, more than half (53.7%) of the female young adults and about two in five (38.4%) of the male young adults reported mild-to-severe depressive symptoms, while almost half of the female young adults (46.7%) and one-third (33.1%) of male young adults reported mild-to-severe anxiety symptoms. Around 21.8% of the young adults met the criteria for at least one of the mental health problems (at least moderate symptoms of depression, GAD, or screening positive for ADHD). The most highly prevalent symptoms of mental health problems were depression symptoms (18.6% for women and 12.7% for men), followed by GAD symptoms (14.4% for women and 9.9% for men) and ADHD-related symptoms (9.7% for women and 7.8% for men). The symptoms of all three mental health problems were more prevalent among women than men. In addition, around one-fifth of young adults reported risky alcohol use monthly and 7.6% weekly risky alcohol use. In contrast to the symptoms of mental health disorders, risky alcohol use is more prevalent among men than women. In general, these results are quite similar to the results from 2018. Compared to the results in 2018, the “severe” symptoms of depression and GAD during the lockdown increased slightly in men (from 10.6% to 12.7% for depression and 8.0% to 9.9% for anxiety) and decreased slightly in women (from 20.3% to 18.6% for depression and 14.7% to 14.4%), whereas ADHD symptoms increased slightly for both men and women. Notably, monthly risky alcohol use decreased dramatically during the lockdown (from 33.7% to 20.8%). This decrease was more pronounced in women than in men.

Overall, more than one-third of the children and adolescents screened positive for one of the mental health problems assessed in our study. The highest prevalence rates were found for ADHD-related symptoms with rates of 22.7% for girls and 23.3% for boys, followed by ODD-related symptoms with rates of 18.2% for girls and 11.2% for boys and anxiety symptoms with rates of 13.6% for girls and 12.5% for boys. Finally, the prevalence rates of depression symptoms were 9.7% among girls and 4.6% among boys. Around 6.0% of girls and 3.3% of boys reported sleeping problems (Table 4).

Among the parents, about a quarter had light-to-severe depression (25.6%) and GAD symptoms (26.5%) (Table 5). These symptoms were more prevalent among women than men. Furthermore, less than 7.0% of mothers and 6.0% of fathers met the criteria for ADHD-related symptoms. In total, 14.5% of mothers and 11.3% of fathers met the criteria for one of the mental health problems listed above.

### 3.3. Internet Use

We report the median of average time spent on the internet per day because the responses had a right-skewed distribution. For both children/adolescents and young adults, the median time spent on the internet per day was 240 min during the COVID-19 lockdown. More than 40% of males and 35% of females used the internet for more than 4 h per day. In addition, more than 10.0% of male young adults used the internet for more than 6 h per day. We further assessed the prevalence of problematic internet use. Around one-third of children/adolescents and one-fifth of young adults met the criteria for problematic internet use. In children, the prevalence was higher among boys than girls; however, females had a higher prevalence of problematic internet use than males among young adults. The most frequently reported problems were “difficult to stop” (loss of control) (24.4%) among girls and “looking forward to the next internet session” (preoccupation) (29.0%) among boys, while the problems that were most frequently reported by young adults of both genders were “difficult to stop” (24.3%) and “go on the internet while feeling down” (coping) (21.2%) (Table 6).

## 4. Discussion

From a large representative sample of young adults, as well as children, adolescents, and their parents, our study found that the most perceived stresses during the first lockdown due to COVID-19 resulted from the disruption of social life and important activities, uncertainty about how long the state of affairs would last, and the pandemic itself (i.e., fear of virus infection), along with distressing news about the pandemic. Children and adolescents felt most stressed by not being able to participate in social activities and normal routines and important plans or events being canceled or postponed. Social activities and relationships with other family members (e.g., grandparents) and with friends and peers are essential for children and adolescents. It is important to help children and adolescents to maintain social contact within their environments with loved ones, friends, schools, and activities. Adults felt most stressed by uncertainty, such as not knowing when the COVID-19 pandemic will end. The health authority could play an important role in reducing anxiety caused by uncertainty and conflicting information or fake news by regularly informing and communicating with the public about the actual situation and clearly explaining the necessary measures and restrictions. Compared to many other European countries, Switzerland’s health system was not overwhelmed due to COVID-19. The government could thus afford to avoid very strict “stay home” curfews, and grocery shopping was allowed. Thus, stress due to trouble obtaining medical care or mental health services or due to trouble obtaining groceries or other needed supplies was less prevalent than in other countries.

Around one-fifth of the young adults met the criteria for one of the mental health problems (depression, GAD, or ADHD-related symptoms), while one-third of children/adolescents screened positive for one of the mental health problems (depression, anxiety, ADHD- or ODD-related symptoms) during the first COVID-19 lockdown in Switzerland. In general, females reported more symptoms of mental health problems and perceived stress than males. These findings are similar to previous studies on university students in Poland and France [32,33]. It is worth noting that the prevalence of mental health problems among young adults during the lockdown remained similar to those before, with the exception that depression symptoms increased slightly among male young adults during the lockdown. Similarly, a repeated survey conducted in Poland with more than 7000 university students showed that depression increased after the lockdown, but not for stress, anxiety, and suicidality [33].

Interestingly, the prevalence of monthly risky alcohol use decreased by about 10% during the lockdown. Young people engage in risky alcohol use more frequently during social events and parties. Therefore, the strict measures during the lockdown, such as social distancing, limited numbers of individuals allowed for private gatherings, and the closing of restaurants and bars, likely contributed to the large decrease in risky monthly binge drinking.

Although it is alarming that one-third of children and adolescents screened positive for one of the mental health problems, these results should be interpreted with caution. First, we used screening instruments that are not intended to establish a diagnosis, but rather to screen mental disorder symptoms in a stepped approach. It is also interesting to note that the most frequent mental health problems among children and adolescents were ADHD-related symptoms, followed by ODD, anxiety, and depression, whereas the mental health problems that were most frequently reported by young adults were depressive symptoms, followed by generalized anxiety and ADHD-related symptoms. Our findings are consistent with the trajectory of these mental health problems found in the literature. Externalizing behavior problems, such as ODD or ADHD, are conceptualized as childhood disorders and often have their onset in early childhood [34,35]. Although these symptoms can be persistent into adulthood, they generally decline in severity from childhood to adulthood [36,37,38]. On the other hand, depression and anxiety often have their onset in adolescence or adulthood, and their prevalence is higher in adults than in children and adolescents [39,40].

Surprisingly, in contrast to practically all previous findings, female children and adolescents reported more ADHD- and ODD-related symptoms than males [41,42,43,44]. One reason might be that females in the present study reported more symptoms overall. Another reason could be that the screening items are not as specific as a complete questionnaire. In general, the lower the age, the more often depressive symptoms are unspecific and are more often expressed with unruly, angry, and aggressive behaviors. With age and the onset of puberty, core depressive symptoms replace these more externally oriented feelings and reactions. It might also be possible that the female children and adolescents expressed internal and external dysfunctional behaviors more often under COVID-19 and more often than males.

The prevalence rates of depression symptoms of 9.7% in girls and 4.6% in boys were lower than the prevalence rates reported in recent reviews that found depression prevalence rates ranging from 22.6 to 43.7% in children and adolescents during the COVID-19 pandemic [11,45]. The girls’ and boys’ prevalence rates of anxiety symptoms of 13.6% and 12.5%, respectively, were also lower than the ones found in children and adolescents examined during the COVID-19 pandemic. These prevalence rates ranged from 18.9 to 37.4% in two recent reviews [11,45]. However, the higher prevalence rates found in these studies could be partly due to the different definitions and instruments used in these studies. Moreover, these studies were conducted mostly in China and Italy [11,45]. These countries experienced more severe circumstances due to the epidemic in terms of cases and mortality and, therefore, had more strict measures imposed by these governments compared to Switzerland. In a general population sample of German children and adolescents, the prevalence rate of generalized anxiety was 24.1% during the COVID-19 pandemic [46]. The same study reported a prevalence rate of 14.9% in another general population sample of German children and adolescents examined before the pandemic, a rate that is similar to the prevalence rates that we found.

In contrast, the ADHD prevalence rates of 22.7% in girls and 23.3% in boys were higher than expected as compared to usual ADHD prevalence rates [41,47,48]. These findings were in line with the finding of elevated hyperactivity symptoms in German children and adolescents during the COVID-19 pandemic [46] and hold true, even when comparing them with the age groups with the highest ADHD prevalence [42] and with prevalence rates of subthreshold ADHD [49,50]. Furthermore, girls usually have a lower ADHD prevalence rate [41,42,43], whereas, in our sample, the prevalence rates of girls and boys were almost identical.

With prevalence rates of 18.2% and 11.2% among girls and boys, respectively, ODD was more prevalent among girls. This runs contrary to the usual finding that, at least in Western countries, ODD is more prevalent among boys than girls [44]. Furthermore, these prevalence rates are higher than what is usually reported, especially for girls [44,51].

Despite the difficulties in comparing prevalence rates from different study samples, the above-cited literature suggests that the prevalence of depression and anxiety symptoms in our children and adolescent sample was lower than expected, whereas ADHD and ODD prevalence rates seemed to be elevated, especially among girls. This resonates with some of the previous research about the impact of the first wave of the COVID-19 pandemic. For example, a study of Italian children found evidence for an increase in hyperactivity/inattention, but not for a change in general psychological adjustment [52]. Among German children and adolescents, no evidence was found for an increase in depressive symptoms, although evidence for an increase in conduct problems and hyperactivity was found [42]. Furthermore, the prevalence of hyperactivity symptoms was slightly higher than the prevalence of general emotional problems during the pandemic, whereas, before the pandemic, this rank order was reversed. Finally, in a longitudinal study of U.S. adolescents, while anxiety, depression, inattention, and oppositionality/defiance symptoms all increased from pre-COVID-19 to spring 2020, only inattention was still elevated by summer 2020 [53]. It might be possible, then, that the stress inflicted by the first wave of the COVID-19 pandemic manifested itself more systematically in externalizing problems, at least in Swiss children and adolescents. This could also explain why girls were found to have similar or higher prevalence rates of ADHD and ODD since they reported higher levels of perceived COVID-19-related stress. It is telling to take a closer look at girls’ most prevalent symptoms across disorders in this context (Table 4), namely, general worry (anxiety), being easily distracted (ADHD), and getting easily upset and losing one’s temper (ODD). Note also that the worry symptom had a much higher prevalence than the anxiety disorder itself. These symptoms could reflect an underlying state of increased stress that resulted in concentration problems, irritability, and worries, and may later manifest in internalizing disorders as well.

Our study found that the median amount of time youths spent on the internet per day was four hours during the first lockdown. It would have been more informative if we had been able to assess the specific types of internet use (time spent on social media, shopping, gaming, etc.). However, due to the time burden of the online survey on participants, we were not able to assess the specific types of internet use. Necessary measures used by many countries to contain the virus spread, such as physical distancing, lockdown, homeschooling, and home office use, inevitably encouraged the public to utilize digital technology for virtually all daily activities, making working and studying remotely possible during the COVID-19 pandemic [54]. In particular, due to the physical distance and lockdown, use of the internet enabled individuals to access reliable information, shop, and keep in contact with friends and family. Our study showed that about 13.8% of children and adolescents, especially girls (18.6% vs. 9.2% boys), and 21.2% of young adults reported using the internet when they were feeling down. On the other hand, previous studies have shown that problematic internet use could have negative consequences on mental health and well-being [55,56,57]. Our preliminary analyses (results not shown) also show that problematic internet use is significantly associated with all mental health problems (symptoms of ADHD, GAD, and depression (odds ratios ranging from 3.1–3.2 in children and 4.6–4.7 in young adults)). Further analyses are necessary to examine the complex associations between perceived stress, problematic internet use, and mental health problems. The present study showed that about 30.0% of children and adolescents and one-fifth of young adults met the criteria for problematic internet use. These results are comparable with two other studies with adolescent samples (24.0% and 33.0%) [55,58]. Since the use of the internet and digital technology will continue to grow, it is important to provide young people with practical guidance to prevent problematic internet use [54].

### Strengths and Limitations

The present study used large representative samples for children/adolescents and their parents, as well as young adults that included respondents from all three language regions in Switzerland, in contrast to many other studies that used convenience or snowball sampling. In addition, we resurveyed the young adults from S-YESMH, which allowed us to compare the participants’ mental health status before and after the lockdown. However, the study was subject to several limitations. First, our survey was conducted 1–3 months after the lockdown due to the inevitable delay of obtaining funding and ethical approval. Therefore, the results might be subject to recall bias. However, our results are quite comparable to other studies. In addition, our results are confined to the effects of the first pandemic wave. They do not capture the effects of the second pandemic wave that gained momentum in October 2020. Further studies should be conducted to monitor the effects of the second lockdown. Second, our survey was conducted online instead of in person, and the instruments we used measured symptoms of mental health problems and cannot be used to establish a diagnosis. In addition, we used the GAD-7, which measures symptoms of generalized anxiety disorder, not anxiety in general. We thus do not have information about the symptoms of other types of anxiety. Finally, non-Swiss residents were underrepresented in our study, which is a reality that is inevitable in most surveys. However, we compared the prevalence of mental disorder symptoms between Swiss and non-Swiss children and found no significant differences (depressive symptoms: 6.7% vs. 6.6%, *p* = 0.99; anxiety symptoms: 12.7% vs. 16.0%, *p* = 0.41; ODD: 14.6% vs. 18.7%, *p* = 0.334; ADHD: 23.0% vs. 24.0%, *p* = 0.838; problematic internet use: 29.6% vs. 25.3%, *p* = 0.433). Furthermore, we cannot rule out the possibility of non-participation bias. As stated in the data analysis section, we found a slightly higher proportion of those who had mental disorder symptoms in 2018 among the non-respondents. In particular, participants in the first survey that had symptoms of depression were less likely to participate in the second survey. In addition, males and non-Swiss participants in the first survey were also less likely to respond in the second survey. Therefore, the prevalence of symptoms of mental health problems could be underestimated. Furthermore, caution should be taken when generalizing our results to other countries due to the different pandemic, economic, and political situations. Within Europe, Switzerland was hit early and comparably hard with the first confirmed case reported on 24 February 2020. However, the health system was not overwhelmed, and the economic contraction was buffered by the government’s economic policy response. As such, the worst-case scenarios did not materialize in Switzerland, and this likely led to a less significant negative impact on mental health.

## 5. Conclusions

With a large representative sample of youths, our study showed considerable stresses that were perceived by young adults, especially among females, during the first COVID-19 pandemic lockdown in Switzerland. Children and adolescents were most stressed by the disruption of social life and important activities/events, whereas adults were most stressed by the uncertainty of the pandemic and the disease itself. Regular information provided by the health authority about the pandemic could help to ease the public’s uncertainty and anxiety. In addition, parents and schools could organize activities to help children and adolescents maintain routines, schedules, social contacts, and support during the lockdown. Therefore, the health and education authorities should provide more support to institutions, such as kindergartens, schools, and sports organizations, to avoid the detriments of longer homeschooling phases, the loss of opportunities to meet peers, and familiar daily routines.

## Figures and Tables

**Table 1 ijerph-18-04668-t001:** Sociodemographic characteristics of the study participants.

Characteristics	Sample 1: Young Adults (*n* = 1627)		Sample 2: Children/Adolescents (*n* = 1146)			Sample 3: Parents (*n* = 1146)
	*n* (%)	Weighted %	*n* (%)		Weighted %	*n* (%)
Gender						
*Male*	638 (39.2)	49.8	569 (49.7)		51.4	382 (33.3)
*Female*	983 (60.4)	49.9	577 (50.4)		48.6	764 (66.7)
*Other*	6 (0.4)	0.3				
Age (mean ± SD)	21.5 ± 1.5 y	21.5 ± 1.7 y		14.3 ± 1.5 y	14.5 ± 1.7 y	47 ± 4.7 y
*19 y*	140 (8.6)	8.7	12 y	203 (17.7)	16.9	Age range 32–55 y
*20 y*	363 (22.3)	23.8	13 y	206 (18.0)	16.7	
*21 y*	345 (21.2)	22.1	14 y	213 (18.6)	16.7	
*22 y*	304 (18.7)	18.2	15 y	224 (19.5)	16.7	
*23 y*	297 (18.3)	16.5	16 y	198 (17.3)	16.4	
*24 y*	178 (10.9)	9.4	17 y	102 (8.9)	16.7	
Nationality						
*Swiss*	1443 (88.7)	81.4		1071 (93.5)	94.0	1038 (90.6)
*Non-Swiss*	184 (11.3)	18.6		75 (6.5)	6.0	108 (9.4)
Education						
*Mandatory and other*	121 (7.4)	7.3		797 (69.6)	65.1	34 (3.0)
*Vocational school 1–2 y*	506 (31.1)	32.5		140 (12.2)	15.7	492 (42.9)
*Vocational school 3–4 y*	763 (46.9)	45.8		209 (18.2)	19.1	193 (16.8)
*College or university*	237 (14.6)	14.4				427 (37.3)
Employment						
*Full time*	457 (28.1)	28.5				510 (44.5)
*Part-time/in education*	1029 (63.2)	62.4				463 (40.4)
*Unemployed*	52 (3.2)	3.0				17 (1.5)
*Other* ^1^	42 (2.6)	2.6				156 (13.6)

^1^: Other includes “in education/military service,” “on welfare,” or “housewife/houseman.”

**Table 2 ijerph-18-04668-t002:** Prevalence of perceived stress due to COVID-19 (%, rank order).

	Young Adults		Children		Parents	
Prevalence of Somewhat/Very in % (Prevalence Rank Order within Group)	Females	Males	Girls	Boys	Mothers	Fathers
1. My family has experienced financial problems	11.3 (13)	9.4 (10)	6.7 (13)	7.0 (11)	12.6 (12)	10.7 (10)
2. Unable to spend time in person with my friends or family	55.2 (1)	43.3 (2)	39.4 (3)	33.3 (3)	38.5 (2)	21.7 (4)
3. Unable to participate in social activities and normal routines	43.2 (4)	37.5 (3)	41.0 (2)	37.9 (1)	25.5 (7)	21.5 (6)
4. Having to change, postpone, or cancel important plans or events	48.8 (3)	44.1 (1)	48.1 (1)	35.1 (2)	33.6 (5)	33.5 (1)
5. Challenges at home or with others	21.8 (9)	13.7 (9)	12.2 (10)	7.9 (10)	13.5 (11)	8.9 (12)
6. My family has experienced trouble getting groceries or other needed supplies	5.7 (14)	4.6 (14)	4.4 (14)	2.6 (14)	7.3 (14)	8.4 (13)
7. Watching or hearing distressing news reports about COVID-19	32.9 (7)	18.9 (7)	24.6 (6)	14.5 (8)	35.6 (3)	24.6 (3)
8. Not being sure about myself or someone close to me getting COVID-19	34.1 (6)	22.5 (6)	23.7 (7)	17.7 (6)	30.5 (6)	21.7 (4)
9. Myself or someone close to me having symptoms or being diagnosed with COVID-19	34.4 (5)	27.4 (5)	25.9 (5)	18.7 (5)	33.8 (4)	19.6 (7)
10. Trouble getting medical care or mental health services	13.2 (11)	8.2 (11)	10.3 (11)	6.9 (12)	15.4 (9)	10.5 (11)
11. Not being sure about when COVID-19 will end or what will happen in the future	53.9 (2)	35.4 (4)	35.0 (4)	23.4 (4)	41.0 (1)	28.0 (2)
12. Difficulty completing my school/work responsibilities online	14.5 (10)	7.7 (13)	18.7 (8)	16.5 (7)	11.6 (13)	8.4 (13)
13. Unable to complete educational or work requirements	24.1 (8)	17.8 (8)	16.9 (9)	11.8 (9)	15.1 (10)	11.4 (9)
14. Needing to take on greater family and/or work responsibilities	13.0 (12)	7.9 (12)	7.5 (12)	5.8 (13)	23.0 (8)	12.3 (8)

**Table 3 ijerph-18-04668-t003:** Prevalence of mental health outcomes among young adults.

	S-YESMH in 2018			During the Lockdown		
Mental Health Problems	Total (%) [95% CI]	Females (%) [95% CI]	Males (%) [95% CI]	Total (%) [95% CI]	Females (%) [95% CI]	Males (%) [95% CI]
Depression ^1^	15.4 [13.6, 17.3]	20.3 [17.6, 23.0]	10.6 [8.1, 13.2]	15.6 [13.7,17.6]	18.6 [15.9,21.2]	12.7 [9.8,15.6]
*No or minimal*	54.3 [51.7, 56.9]	46.4 [43.1, 49.6]	62.3 [58.1, 66.4]	54.0 [51.3, 56.7]	46.3 [42.9, 49.7]	61.6 [57.2, 65.9]
*Mild*	30.0 [27.7, 32.4]	35.2 [32.1, 38.3]	24.9 [21.3, 28.4]	30.4 [27.8, 33.0]	35.1 [31.8, 38.4]	25.7 [21.7, 29.8]
*Moderate*	11.0 [9.3, 12.7]	12.3 [10.1, 14.5]	9.7 [7.0, 12.3]	11.0 [9.2, 12.7]	12.2 [10.0, 14.4]	9.8 [7.1, 12.5]
*Severe*	4.6 [3.6, 5.7]	6.1 [4.6, 7.6]	3.2 [1.8, 4.6]	4.7 [3.6, 5.7]	6.4 [4.7, 8.1]	2.9 [1.6, 4.2]
Generalized anxiety disorder ^1^	11.4 [9.8, 13.0]	14.7 [12.4, 17.0]	8.0 [5.9, 10.2]	12.2 [10.4, 14.0]	14.4 [11.9, 17.0]	9.9 [7.3, 12.5]
*No or minimal*	61.6 [59.1, 64.0]	54.5 [51.3, 57.8]	68.6 [64.9, 72.3]	60.1 [57.5, 62.8]	53.4 [50.0, 56.7]	66.9 [62.7, 71.1]
*Mild*	26.5 [24.3, 28.8]	31.5 [28.4, 34.5]	21.6 [18.4, 24.9]	27.7 [25.2, 30.3]	32.2 [28.9, 35.5]	23.2 [19.3, 27.2]
*Moderate*	8.4 [7.0, 9.8]	9.8 [7.9, 11.8]	6.9 [4.9, 9.0]	8.6 [7.0, 10.1]	10.2 [8.0, 12.4]	7.0 [4.8, 9.1]
*Severe*	3.5 [2.6, 4.4]	4.2 [2.9, 5.4]	2.9 [1.5, 4.2]	3.6 [2.6, 4.6]	4.3 [2.9, 5.7]	2.9 [1.4, 4.5]
ADHD-related symptoms	8.3 [6.7, 9.8]	9.1 [7.2, 11.0]	7.4 [5.1, 9.8]	8.8 [7.3, 10.3]	9.7 [7.8, 11.6]	7.8 [5.6, 10.1]
Any mental health problems	23.0 [20.8, 25.2]	27.4 [24.4, 30.3]	18.6 [15.4, 21.9]	21.8 [19.6, 24.1]	25.2 [22.2, 28.2]	18.5 [15.1, 21.8]
Risky alcohol use (at least monthly)	33.7 [31.3, 36.2]	33.5 [30.6, 36.4]	34.0 [30.1, 37.9]	20.8 [18.3, 23.3]	18.3 [15.3, 21.3]	23.1 [19.1, 27.0]
Risky alcohol use (at least weekly)	9.8 [8.2, 11.3]	8.3 [6.6, 10.1]	11.2 [8.7, 13.7]	7.6 [5.9, 9.2]	6.0 [4.1, 8.0]	9.0 [6.3, 11.7]

^1^: Percentage for “moderate” and “severe.”

**Table 4 ijerph-18-04668-t004:** Prevalence of mental health problems for children and adolescents.

Mental Health Status	Total (%) [95% CI]	Girls [95% CI]	Boys [95% CI]
Depression	7.1 [5.5,9.1]	9.7 [7.3,12.8]	4.6 [2.8,7.5]
1. Feeling down, depressed, irritable, or hopeless (PHQ-2)	2.0 [1.3,3.2]	3.3 [2.0,5.4]	0.8 [0.3,2.4]
2. Little interest or pleasure in doing things (PHQ-2)	1.5 [0.9,2.5]	1.5 [0.7,3.1]	1.5 [0.7,3.0]
3. Trouble falling asleep, staying asleep, or sleeping too much (sleep problem)	4.6 [3.4,6.2]	6.0 [4.2,8.5]	3.3 [2.0,5.5]
Anxiety ^1^	13.1 [11.0,15.5]	13.6 [10.8,17.1]	12.5 [9.8,16.0]
1. Worry about things	16.4 [14.2,19.0]	21.3 [17.8,25.4]	11.8 [9.2,15.1]
2. Feel afraid	5.9 [4.5,7.7]	9.0 [6.6,12.2]	3.0 [1.8,4.8]
3. Worry about being away from my parents	7.6 [6.1,9.5]	9.0 [6.8,12.0]	6.3 [4.4,8.9]
4. Feel scared if I have to sleep on my own	3.2 [2.3,4.6]	4.2 [2.6,6.7]	2.4 [1.4,4.0]
5. Have trouble going to school in the mornings because I feel nervous or afraid	4.7 [3.4,6.4]	5.3 [3.7,7.5]	4.0 [2.3,6.9]
6. Suddenly start to tremble or shake when there is no reason for this	4.3 [3.2,5.8]	6.2 [4.3,8.9]	2.5 [1.4,4.3]
7. Worry that I will suddenly get a scared feeling when there is nothing to be afraid of	4.5 [3.3,6.0]	6.8 [4.8,9.6]	2.2 [1.2,4.2]
8. Feel scared if I had to stay away from home overnight	4.0 [2.8,5.7]	4.1 [2.7,6.3]	3.9 [2.3,6.7]
ADHD-related symptoms	23.0 [20.4,25.9]	22.7 [19.0,26.9]	23.3 [19.7,27.3]
1. Difficult to sustain attention on tasks (e.g., homework) or play activities (e.g., a board game)	9.1 [7.5,11.1]	8.5 [6.3,11.4]	9.8 [7.5,12.7]
2. Being easily distracted during tasks that require attention	13.7 [11.7,16.0]	13.4 [10.6,16.9]	14.0 [11.2,17.2]
3. Difficult to stay seated when you are expected to	6.6 [5.3,8.3]	6.0 [4.2,8.5]	7.2 [5.3,9.6]
4. Act impulsively without thinking about consequences	8.1 [6.4,10.2]	8.9 [6.5,12.0]	7.4 [5.2,10.5]
ODD-related symptoms	14.6 [12.5,17.1]	18.2 [15.0,22.1]	11.2 [8.6,14.5]
1. Easily get upset and lose your temper	9.4 [7.7,11.4]	12.3 [9.6,15.7]	6.6 [4.8,9.1]
2. Often argue and talk back with your parents or teachers	6.3 [4.9,8.2]	7.7 [5.6,10.5]	5.1 [3.3,7.7]
3. Defy or disobey rules at home, school, or other places	4.8 [3.6,6.4]	5.1 [3.3,7.9]	4.5 [3.0,6.7]
Any mental health problems	35.2 [32.1,38.4]	39.1 [34.7,43.7]	31.5 [27.3,36.0]

^1^: Cutoff score was 7.5 for girls and 5.5 for boys based on the literature.

**Table 5 ijerph-18-04668-t005:** Prevalence of mental health problems among parents.

Mental Health Problems (% [95% CI])	Total	Mothers	Fathers
Depression ^1^	7.6 [6.2,9.3]	8.1 [6.4,10.3]	6.5 [4.5,9.5]
*No or minimal*	74.4 [71.8,76.9]	71.7 [68.4,74.8]	79.8 [75.5,83.6]
*Light*	18.0 [15.6,20.3]	20.2 [17.5,23.2]	13.6 [10.5,17.4]
*Moderate*	5.0 [3.9,6.4]	5.6 [4.2,7.5]	3.7 [2.2,6.1]
*Severe*	2.6 [1.8,3.7]	2.5 [1.6,3.9]	2.9 [1.6,5.1]
Generalized anxiety disorder ^1^	6.6 [5.3,8.2]	6.5 [5.0,8.5]	6.8 [4.7,9.8]
*No or minimal*	73.6 [70.9,76.0]	70.4 [67.1,73.6]	79.8 [75.5,83.6]
*Light*	19.8 [17.6,22.2]	23.0 [20.2,26.2]	13.4 [10.3,17.1]
*Moderate*	4.5 [3.4,5.8]	4.6 [3.3,6.3]	4.2 [2.6,6.7]
*Severe*	2.2 [1.5,3.2]	2.0 [1.2,3.2]	2.6 [1.4,4.8]
ADHD-related symptoms	5.8 [4.6,7.4]	6.2 [4.7,8.1]	5.2 [3.4,8.0]
Any mental health problems	13.4 [11.6,15.5]	14.5 [12.2,17.2]	11.3 [8.5,14.8]

^1^: % for “moderate” and “severe.”

**Table 6 ijerph-18-04668-t006:** Internet use during the lockdown due to COVID-19.

	Children/Adolescents			Young Adults		
	Total	Girls	Boys	Total [IQR]	Females	Males
Average minutes per day median [IQR]	240 [120,339]	180 [120,300]	240 [120,360]	240 [120,301]	210 [120,300]	240 [147,360]
Average minutes per day % [95% CI]						
0–120 min	29.0 [26.0,32.1]	32.2 [28.1,36.5]	26.0 [21.8,30.6]	25.8 [23.5, 28.2]	27.7 [24.6, 30.9]	24.0 [20.4, 27.5]
120–240 min	32.7 [29.7,35.8]	32.4 [28.2,36.9]	32.9 [28.7,37.5]	35.4 [32.8, 38.0]	36.8 [33.6, 40.0]	34.0 [30.0, 38.1]
241–480 min	30.7 [27.8,33.9]	27.2 [23.4,31.5]	34.0 [29.7,38.6]	30.3 [27.6, 32.9]	29.4 [26.3, 32.5]	31.2 [26.9, 35.5]
>481 min	7.9 [6.0,9.5]	8.2 [5.9,11.2]	7.1 [5.0,9.8]	8.4 [6.8, 10.0]	6.1 [4.4, 7.8]	10.8 [8.0, 13.5]
Problem internet use % [95% CI]	30.1 [27.2,33.2]	28.7 [24.7,33.0]	31.5 [27.3,36.0]	21.3 [18.9, 23.7]	22.6 [19.7, 25.4]	20.0 [16.1, 23.9]
1. Difficult to stop using the internet (loss of control)	25.5 [22.8,28.4]	24.4 [20.8,28.5]	26.5 [22.6,30.8]	24.3 [21.8, 26.8]	27.3 [24.3, 30.3]	21.3 [17.3, 25.2]
2. Prefer to use the internet instead of spending time with others (preoccupation)	18.4 [16.0,21.0]	18.7 [15.3,22.6]	18.1 [15.0,21.8]	8.8 [7.3, 10.3]	8.8 [6.8, 10.8]	8.8 [6.6, 11.0]
3. Shortage of sleep because of the internet (loss of control)	7.0 [5.5,8.9]	7.5 [5.4,10.4]	6.5 [4.6,9.2]	9.5 [7.6, 11.5]	8.6 [6.7, 10.5]	10.5 [7.1, 14.0]
4. Look forward to your next internet session (preoccupation)	22.5 [19.9,25.4]	15.7 [12.7,19.3]	29.0 [24.9,33.4]	8.5 [7.0, 10.0]	7.8 [6.0, 9.6]	9.2 [6.9, 11.5]
5. Unsuccessfully tried to spend less time on the internet (loss of control)	7.7 [6.1,9.6]	9.8 [7.3,13.0]	5.7 [4.0,8.0]	8.2 [6.8, 9.7]	9.0 [7.2, 10.8]	7.5 [5.3, 9.7]
6. Neglect your daily obligations because you prefer to go on the internet (conflict)	7.2 [5.7,9.0]	6.6 [4.6,9.3]	7.8 [5.7,10.5]	6.0 [4.3, 7.8]	4.7 [3.3, 6.0]	7.4 [4.2, 10.5]
7. Go on the internet when you are feeling down (coping)	13.8 [11.7,16.2]	18.6 [15.3,22.5]	9.2 [6.8,12.2]	21.2 [18.7, 23.6]	22.8 [19.8, 25.7]	19.6 [15.6, 23.5]
8. Feel restless, frustrated, or irritated when you cannot use the internet (withdrawal symptoms)	10.4 [8.6,12.5]	10.5 [8.0,13.6]	10.3 [7.9,13.3]	4.3 [3.1, 5.5]	4.4 [2.8, 5.9]	4.2 [2.6, 5.9]

## Data Availability

The datasets analyzed in the current study are not publicly available due to the conditions specified in the data protection contract for this study.

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
