# Peer review of "Stress and Mental Health among Children/Adolescents, Their Parents, and Young Adults during the First COVID-19 Lockdown in Switzerland"

_ijerph, 2021, doi:10.3390/ijerph18094668_

Round 1

Reviewer 1 Report

Thank you for allowing me to review this paper.  This is an interesting topic and is very important globally. Some comments to improve the paper.

-The introduction section can be improved by making the presentation explicit in terms of whether it is referring to the pandemic or not. From lines 47 -63, it is not always very clear under which conditions the cited sources were presented. It is important to understand that the mental health conditions experienced are due to the pandemic or it was already present under other circumstances. Also, a reference is needed for the statement in lines 48 to 50.

-The research design was scientifically sound. It was explained how the data was collected, where and when within the ethical guidelines. However, I feel the authors can make a greater, more interesting contribution by analyzing the data beyond the descriptive statistics presented. What are the relationships (if any) between the variables measured. For example, what is the role of perceived stress on mental health outcomes, or perceived stress on Internet use? The authors have a very good data set, which can be analyzed in more detail to provide a richer discussion of the results presented. As a consequence, the complete discussion section will need to be updated.

-A reference is missing for claim made in line 379-381.

Reviewer 2 Report

Congratulations on the study. I miss a more detailed description of the socioeconomic circumstances surrounding the first lockdown in Switzerland, such as habitability conditions, population density, impact of COVID in that country ... I also think it necessary to include more quality evidence to support the justification of study and discussion. Again congratulations on the job. Hopefully more similar studies will appear that include younger children.

Author Response

Review 2

Comments and Suggestions for Authors

Congratulations on the study. I miss a more detailed description of the socioeconomic circumstances surrounding the first lockdown in Switzerland, such as habitability conditions, population density, impact of COVID in that country ... I also think it necessary to include more quality evidence to support the justification of study and discussion. Again congratulations on the job. Hopefully more similar studies will appear that include younger children.

Responses: Thank you for the reviewer’s encouragement and thoughtful comments that have helped us to improve the manuscript. We have elaborated on these missing points in the discussion section. Please see line 501-508.

Reviewer 3 Report

Thank you for the opportunity to review this manuscript. It focuses on the timely issue of the impact of lockdowns on children and adolescents’ mental health. Overall, the manuscript is well written and clearly structured, however, there are some remaining points which I feel need additional attention. In particular, I feel that the data should be analyzed using more robust statistical methods to better capture possible group differences (rather than merely comparing percentages descriptively). Also, the debatable operationalization of problematic internet use deserves more attention in the introduction and discussion. For more details see my comments below.

ABSTRACT:
- consider specifying the age of included “young adults” already in the abstract for more transparency.
- the sentence “and 1146 children and adolescents ages 12-17 years“ should read “aged 12-17 years”

INTRODUCTION:

- The rationale of Including problematic internet use (PIU) is not clear, also on the basis of past studies which are discussed in the introduction – why was PIU included separately? And not, for instance, Internet Gaming Disorder which – in contrast to PIU – is a diagnosis according to ICD 11? Maybe the authors could elaborate more on this issue in their introduction. Also, the underlying theoretical conception of PIU should be presented.

RESULTS:
- the most important critique regarding the reported results is that only % and CI are reported; all comparisons are based on merely descriptive statistics, no inferential statistics are used; this is all the more surprising, since the sample sizes are large enough. I suggest applying robust statistical methods to analyze group differences.

- PIU: it could be of merit to also analyze the associations between PIU and mental disorders as suggested in the discussion (see: “On the other hand, previous studies have shown that 452

problematic internet use could have negative consequences on mental health and well- 453

being [48-50].”). As far as I can see, the data should be fit to do this. If not, the reason for not exploring this issue with the current data set should be included in the discussion/limitation section.

- consider providing the age range for all the three groups in the text.

- check all decimal places for the reported percentages, e.g. in line 240 the decimal place (0?) is missing

- instead of writing “female parents” I suggest using “mothers”

- demographics: is any other demographic information (apart from nationality) available? – such as educational background, income etc. – everything that would shed more light on the samples’ social backgrounds and hence, help in interpreting the data. If not, this should definitely be explained and included as a major limitation in the discussion/limitations section

DISCUSSION:
- problematic internet use: Again, I am missing a more thorough discussion of the results’ implications, particularly with regards to PIU. For instance, a major problem in the operationalization of PIU is that it is quite unspecific. There are indications, that rather than the use of the internet per se, the usage of specific applications (e.g., social networks, gaming, shopping, video streaming) is more indicative of a problematic use. Here, the internet serves only as a vehicle.

ADDITIONAL COMMENTS

  • The bullet points “internet use” and “perceived stress” should be in italics too (like “Symptoms of mental disorders”)
  • 5 line 216: there is one full stop too much in this sentence: “non-Swiss.”
  • As far as I am informed, SD are per convention reported with “=” not “:”
  • Line 235 – check the spaces; there seem to be a couple too many

Reviewer 4 Report

Dear authors, congratulations on your highly relevant paper based on your excellent research. Please underline a bit more that results cannot be generalized due to the different (political) situations in each country. You are already mentioning that and I think your results might be generalized but one should be very careful - especially with such significant and relevant results. From my viewpoint your paper can be published after adding that and another spellcheck.

Author Response

Reviewer 4

Dear authors, congratulations on your highly relevant paper based on your excellent research. Please underline a bit more that results cannot be generalized due to the different (political) situations in each country. You are already mentioning that and I think your results might be generalized but one should be very careful - especially with such significant and relevant results. From my viewpoint your paper can be published after adding that and another spellcheck.

Response: Thank you very much for the reviewer’s support. We have added a few sentences in the discussion as suggested.

Round 2

Reviewer 3 Report

The authors have considered some of my comments, however, the two most important issues remain:

1) the introduction should at least include some reference to the construct of PIU, in order for the readers to be able to assess its meaning. In contrast to alcohol abuse or depression for which there is a general agreement and inclusion in the according ICD/DSM, PIU is a highly debated construct.

2) I still find it problematic if the authors do not include more complex statistical analysis. As a reader, it is unclear why this was not done since the data quality is obviously sufficient. Also, no rationale is provided for this mode of action; and the addition of only rudimentary analyses in the discussion with the statement that “further analysis are necessary” runs the risk of confusing the readers even more.

Author Response

The editor has agreed that we can waive this part and we address academic Editor's instead.